# Machine Learning Techniques for Canine Myxomatous Mitral Valve Disease Classification: Integrating Anamnesis, Quality of Life Survey, and Physical Examination

**DOI:** 10.3390/vetsci11030118

**Published:** 2024-03-06

**Authors:** Javier Engel-Manchado, José Alberto Montoya-Alonso, Luis Doménech, Oscar Monge-Utrilla, Yamir Reina-Doreste, Jorge Isidoro Matos, Alicia Caro-Vadillo, Laín García-Guasch, José Ignacio Redondo

**Affiliations:** 1Internal Medicine, Veterinary Medicine and Therapeutic Research Group, Faculty of Veterinary Science, Research Institute of Biomedical and Health Sciences (IUIBS), University of Las Palmas de Gran Canaria, 35017 Las Palmas de Gran Canaria, Spain; jengelmanchado@yahoo.es (J.E.-M.); alberto.montoya@ulpgc.es (J.A.M.-A.); laingarcia@gmail.com (L.G.-G.); 2Cardiology Service, AniCura Benipeixcar Veterinary Hospital, 46009 Valencia, Spain; 3Cardiology Service, AniCura San Vicente Veterinary Hospital, 03690 Alicante, Spain; 4Cardiology Service, AniCura San Francisco Veterinary Hospital, 12500 Vinaròs, Spain; 5Department of Mathematics, Physics and Technological Sciences, Higher School of Technical Education, Cardenal Herrera-CEU University, 46115 Valencia, Spain; luis.domenech@uchceu.es; 6Cardiology Service, Mediterráneo Veterinary Hospital, Evidensia IVC, 28007 Madrid, Spain; omongeutrilla@gmail.com; 7Cardiology Service, IVC Evidensia, Los Tarahales Veterinary Hospital, 35019 Las Palmas de Gran Canaria, Spain; yamir.reina@ivcevidensia.es; 8Department of Animal Medicine and Surgery, Faculty of Veterinary Medicine, Complutense University, 28040 Madrid, Spain; 9Cardiology & Respiratory Service, IVC Evidensia Molins Veterinary Hospital, 08620 Barcelona, Spain; 10Cardiology Service, IVC Evidensia del Mar Veterinary Hospital, 08005 Barcelona, Spain; 11Department of Animal Medicine and Surgery, Faculty of Veterinary Medicine, Cardenal Herrera-CEU University, 46115 Valencia, Spain; nacho@uchceu.es

**Keywords:** anamnesis, clinical diagnosis, machine learning, predictive model, myxomatous mitral valve disease, dog

## Abstract

**Simple Summary:**

Myxomatous mitral valve disease is dogs’ most common acquired heart disease. The gold standard for its definitive diagnosis is echocardiography. This study aimed to develop a tool that uses a quality of life survey, structured anamnesis, and physical examination to predict the American College of Veterinary Internal Medicine classification stages. Accurately identifying a patient’s stage is crucial to evaluating when treatment should be initiated and tailoring it to their ACVIM stage. The study analysed 1011 dogs from 23 hospitals, and the results showed that the majority of patients were successfully classified into the control group (healthy dogs), stage B (dogs with a heart murmur but are asymptomatic), and stage C (dogs with heart failure). However, efficient results were not obtained to differentiate between stage B1 (dogs with a heart murmur and without heart enlargement) and stage B2 (dogs with a heart murmur and heart enlargement). Further studies should be carried out to implement these techniques and improve their diagnostic value in veterinary cardiology.

**Abstract:**

Myxomatous mitral valve disease (MMVD) is a prevalent canine cardiac disease typically diagnosed and classified using echocardiography. However, accessibility to this technique can be limited in first-opinion clinics. This study aimed to determine if machine learning techniques can classify MMVD according to the ACVIM classification (B1, B2, C, and D) through a structured anamnesis, quality of life survey, and physical examination. This report encompassed 23 veterinary hospitals and assessed 1011 dogs for MMVD using the FETCH-Q quality of life survey, clinical history, physical examination, and basic echocardiography. Employing a classification tree and a random forest analysis, the complex model accurately identified 96.9% of control group dogs, 49.8% of B1, 62.2% of B2, 77.2% of C, and 7.7% of D cases. To enhance clinical utility, a simplified model grouping B1 and B2 and C and D into categories B and CD improved accuracy rates to 90.8% for stage B, 73.4% for stages CD, and 93.8% for the control group. In conclusion, the current machine-learning technique was able to stage healthy dogs and dogs with MMVD classified into stages B and CD in the majority of dogs using quality of life surveys, medical history, and physical examinations. However, the technique faces difficulties differentiating between stages B1 and B2 and determining between advanced stages of the disease.

## 1. Introduction

Myxomatous mitral valve disease (MMVD) is the most common heart disease in dogs [1,2,3]. It accounts for up to 75% of all cardiovascular diseases in dogs, with an exceptionally high prevalence in senior and small dog breeds, such as Cavalier King Charles Spaniels (CKCS) [4,5]. MMVD significantly reduces life expectancy and quality of life in affected dogs [6,7]. Early diagnosis and staging of this condition are essential to determining the appropriate time to start therapy, achieving a better prognosis in most dogs [8].

The American College of Veterinary Internal Medicine (ACVIM) developed a classification system (stages A, B, C, and D) for MMVD, emphasising the importance of identifying the disease’s severity and response to treatment [9]. Accurate and timely diagnosis is typically based on a combination of ancillary tests, including thoracic radiography, electrocardiography, and blood tests. Echocardiography is the most important clinical test to confirm a definitive diagnosis [9,10]. However, it is not possible to classify patients solely based on medical history and clinical signs, leading to misdiagnosis, especially when other non-cardiac illnesses present similar signs [11,12,13]. 

Although highly effective in diagnosing MMVD and its progression, echocardiography is only sometimes readily available due to the need for specialised equipment and expertise [14]. Therefore, there is a need for user-friendly tools to assist general veterinarians in classifying MMVD, especially in cases where advanced diagnostic tests are unavailable [15,16] and prompt action is crucial.

Machine learning techniques have gained recognition for their ability to analyse extensive datasets, offering flexibility, and scalability compared to traditional biostatistical methods, making them applicable to many tasks, such as risk stratification, diagnosis, classification, and survival predictions [17]. Human cardiology has successfully used these techniques to aid diagnosis and risk stratification [17,18]. However, their application in veterinary cardiology, especially for patient consultation, is still in its early stages [19,20,21]. A previous study demonstrated the usefulness of quality of life surveys in predicting outcomes in dogs with MMVD [22]. In addition, a recent study has shown the ability of machine learning techniques to classify patients affected by MMVD using thoracic radiographs [21].

The primary aim of this study was to assess the potential of a structured medical history complemented by a quality of life survey and physical examination analysed through machine learning to assist in classifying MMVD at various stages in dogs. Moreover, the purpose was to explore how owners perceive the disease in dogs with MMVD, even when they are unaware of the specific cause behind their pets’ clinical signs.

## 2. Materials and Methods

An observational, prospective, and multicentre clinical study was conducted across twenty-three veterinary hospitals in Spain, Brazil, Argentina, Chile, and Costa Rica. All participating veterinarians had at least five years of experience in veterinary cardiology, further substantiated by postgraduate training in this specialised field (Ph.D. in cardiology research, specialised accreditation in cardiology, International School of Veterinary Postgraduate Studies (ISVPS) recognition, certificate in advanced veterinary cardiology by the RCVS, and cardiology resident by the ACVIM residency programme authorised to perform evaluations). Ethical approval was granted by the Animal Experimentation Ethics Committee of CEU Cardenal Herrera University (Spain) under reference number CEEA 22/06. 

A total of 1011 client-owned dogs were evaluated; 64 healthy dogs were integrated into a control group; and 947 dogs with clinical findings of a left apical systolic murmur, which was confirmed through echocardiography, were integrated into a MMVD group. The inclusion criteria did not discriminate based on sex, breed, reproductive status, or body weight. However, dogs younger than one year old were excluded. The patients’ owners were fully informed about the nature of the study, and their written consent was obtained to use their questionnaire responses and patient examination data for research purposes. The inclusion criteria for both control and MMVD groups required that the owner complete the FETCH-Q quality of life survey and that each dog be evaluated through history, physical examination, and echocardiography.

The control group were animals evaluated prior to elective surgery did not present clinical signs (absence of cardiorespiratory clinical signs, heart murmur, and systemic or organ-related diseases) and did not receive any medication. The MMVD group were dogs with the presence of a left-sided systolic heart murmur and were subsequently classified according to the ACVIM guidelines after echocardiography and radiographic examination (stage B1/B2/C/D). In particular, dogs previously treated or diagnosed with MMVD were excluded from the study, and dogs with other structural cardiovascular disorders (congenital, infectious, or degenerative) were also excluded from the study design. However, the presence of other non-cardiovascular comorbidities was not considered an exclusion factor due to the heterogeneous nature of the sample and the common occurrence of comorbidities in patients with MMVD, along with the degenerative progression of the disease.

At the time of completing the quality of life questionnaire, the owner of the patients in the group with MMVD possessed only the knowledge that their dog exhibited a heart murmur. Similarly, the sonographers conducting the echocardiography were aware that the referral was based on the presence of a heart murmur but lacked information regarding the anamnesis, physical examination, and specific stage according to the ACVIM classification.

A structured consultation comprised four distinct parts: a quality of life survey [7,22], anamnesis, a comprehensive physical examination, and an echocardiography examination, all conducted on the same day and with the same patient. The patient’s medical history was meticulously documented, and the owner was asked about specific clinical signs in the previous two weeks, such as cough, dyspnoea, syncope, exercise intolerance, hypoxia/anorexia, or weight loss. Furthermore, the owner completed the Spanish version of the FETCH-Q quality of life survey [23]. A thorough physical examination encompassed the assessment of heart rate (HR), respiratory rate (RR), and rectal temperature (RT). Dogs were further categorised based on murmur grade, according to the I-VI system [24,25]. Body weight was recorded in kilograms, and the body condition score was assessed on a scale of 1 to 9 [26]. Blood pressure was measured with the following devices (SunTech Vet20, Bbraun Vet 25, and Vet30), according to ACVIM guidelines [27]. Five measurements were taken, and the values of the two extremes were discarded. With the other three values, a mean was obtained. One minute was allowed to elapse between measurements.

To standardise echocardiographic measurements, all investigators possessed extensive sonographer experience and adhered to predefined criteria [28]. Key measurements included the assessment of the left atrium and ascending aorta diameter, enabling the calculation of the left atrium/aorta ratio (LA/Ao). This ratio was determined from a 2D right parasternal short-axis view during early ventricular diastole. Additional measurements included left ventricular internal diameter in diastole (LVIDD) and normalised to body weight (LVIDDN) using the formula: LVIDDN = LVIDD (cm)/weight^0.294 (kg) [29]. The echocardiographic classification of mitral disease was conducted according to the ACVIM criteria [9], categorising patients into stages B1, B2, C, and D. Additionally, the mitral valve insufficiency (MINE score) was assessed [30]. According to the ACVIM guidelines for the classification of MMVD, thoracic radiological studies were performed for the correct diagnosis of the animals classified in stages C and D [9].

Echocardiographic data were collected using specialised veterinary cardiology equipment equipped with appropriate probes and software [Philips Affinity 50C, (Amsterdam, Netherlands); General Electric Vivid Iq, (Boston, MA, USA); Mindray Animalcare Vetus 7, (Shenzhen, China); and M8 and Canon a450, (Tokyo, Japan); with phased array probes between 2.5 and 12 MHz), and a uniform protocol was followed for image acquisition. Images were subsequently reviewed by the lead author (JEM) and a board-eligible American College of Veterinary Cardiology (YRD) member, with any substandard images being excluded from analysis. 

Statistical analysis was performed using the R software (version 4.3.0, R Core Team, 2023, Vienna, Austria). Descriptive statistics summarised animal history data and were presented as mean ± SD, the number of observations, and percentages. Responses to the FETCH-Q scale were analysed using the Likert package for R [31] and represented as Likert plots. Univariate analysis was conducted to investigate differences in proportions between categories using the chi-square test [32] and a one-way ANOVA test for quantitative variables, with significance defined at *p* < 0.05.

Classification trees were developed using the rpart function of the rpart statistical package [33] to predict the stage of mitral disease as classified by the ACVIM. This was achieved using three approaches: (1) utilising the FETCH-Q survey, (2) relying on clinical signs identified during the physical examination, and (3) combining the FETCH-Q survey and physical examination findings. A minimum of 100 cases were required for a partition to be performed. The analysis was conducted in two parts: first, for the five ACVIM categories (A, B1, B2, C, and D), and second, a simplified model unifying categories B1 and B2 into classes B, C, and D into category CD. Classification trees were visualised using the rpart. plot function of the rpart.plot package [34].

Furthermore, data were analysed using the random Forest package [35], wherein 66% of the data were used as a learning sample to construct a classification tree, with a minimum of five observations per node. The remaining 33% of the data were treated as out-of-bag data for evaluating the sensitivity and specificity of the classification tree. This process was repeated 1000 times, and the weight of each ACVIM category was adjusted based on the relative percentage frequency of cases analysed. Finally, sensitivity and specificity were calculated by comparing observed results with those predicted by the classification forests, utilising the caret package for R [36]. 

The authors have thoroughly and comprehensively reviewed the content of the article. Additionally, the Grammarly assistant (standard version, 2023, San Francisco, CA, USA) has scrutinised the writing of the article to ensure effective presentation of information and to prevent spelling and grammatical errors in the English language. 

## 3. Results

The study encompassed 1011 dogs, comprising 482 females and 529 males, with a median age of 12.0 years (range: 1.0 to 19.0 years) and a median body weight of 7.0 kg (range: 1.0 to 48.5 kg). The most represented breeds included crossbreeds (n = 371), Yorkshire terriers (n = 128), Chihuahuas (n = 105), Maltese (n = 59), Poodles (n = 55), and Dachshunds (n = 40), while other breeds accounted for the remaining dogs (n = 253). According to the ACVIM classification criteria, 64 dogs fell into the control group, 273 in B1, 357 in B2, 291 in C, and 26 in D. Table 1 and Table 2 represent some demographic data separated by ACVIM groups.

Figure 1, Figure 2 and Figure 3 illustrate the responses to the FETCH-Q questions across different ACVIM stages. It is noteworthy that, with the exception of Q06, all inquiries demonstrated statistically significant differences between groups. The observed variances imply a marked elevation in response scores with increasing severity according to the ACVIM classification, strongly suggesting a decline in quality of life.

Table 3 and Table 4 provide insights into clinical signs and physical examination findings, respectively, categorised by ACVIM classification. Significant differences emerged in the proportion of various clinical indicators, including COUGH, DYSPNOEA, SYNCOPE, EXERCISE INTOLERANCE, HYPOREXIA/ANOREXIA, WEIGHT LOSS, and MURMUR GRADE. In general, clinical signs exhibited a progressive pattern of aggravation with higher ACVIM stages. Notable differences were also observed in heart rate (HR), respiratory rate (RR), and capillary refill time (CRT). However, no significant differences were noted in systolic blood pressure (SBP), diastolic blood pressure (DBP), and median arterial pressure (MAP).

Figure 4 and Figure 5 present the classification tree and the variable importance plot derived from the random forest analysis for the FETCH-Q survey, respectively. 

Similarly, Figure 6 and Figure 7 depict the classification tree and variable importance plot obtained from the random forest analysis for the anamnesis and the physical examination. 

Figure 8 and Figure 9 extend this analysis to include the FETCH-Q, the anamnesis, and the physical examination.

Table 5 summarises the model’s ability to correctly categorise dogs based on ACVIM classification in the three analyses. Notably, combining the FETCH-Q scale and physical examination significantly improved classification accuracy compared to using each component individually. In the combined model, the overall accuracy reached 0.64 (95% CI: 0.609–0.669; *p* < 0.0001; Kappa’s Cohen: 0.489), whereas individual models yielded accuracies of 0.484 (95% CI: 0.453–0.515; *p* < 0.0001; Kappa’s Cohen: 0.263) for the FETCH-Q scale-only model and 0.599 (95% CI: 0.568–0.630; *p* < 0.0001; Kappa’s Cohen: 0.435) for the clinical signs-only model. In the complex model, 96.9% of the control group category, 49.8% of B1, 62.2% of B2, 77.2% of C, and 7.7% of D were correctly classified. Notably, B1 dogs were often confused with B2 (83.2% of B1 dogs misclassified as B2), and B2 dogs were mistaken for B1 and C (57.0% of B2 dogs misclassified as B1 and 43.0% as C). Dogs in category C were frequently misclassified as B2 (86.3%), and D tended to be confused with C (87.5%).

In the simplified model, which grouped categories B1 and B2 as B and C and D as CD, an improvement in classification accuracy was evident. Specifically, 93.8% of the control group, 90.8% of B, and 73.4% of CD were correctly classified, with 9.2% of B misclassified as CD and 27.0% of CD misclassified as B (Table 6). The accuracy of this model reached 85.5% (95% CI: 0.832–0.877; *p* < 0.0001; Kappa’s Cohen: 0.710). The regression tree and variable importance plot for the simplified model are detailed in Figure 10 and Figure 11.

## 4. Discussion

Traditionally, suspicion of MMVD in dogs has been described by veterinarians based on the presence of a heart murmur and/or cardiorespiratory signs. Echocardiography is the definitive diagnostic method and the preferred technique for categorising the severity of the disease. The developed project has demonstrated adequate results to differentiate between healthy animals and those with MMVD using machine-learning algorithms, which use clinical history, a quality of life survey, and a physical examination. The information obtained can aid in the initial assessment of dogs with MMVD before confirmation by echocardiographic study. Furthermore, the technique allowed for capturing the owner’s perspective on disease progression, contributing to a comprehensive understanding of the disease [16]. 

The complex model correctly classified control dogs and stage C patients, achieving 96.9% and 77.2% accuracy, respectively. However, it needed support to accurately identify the B groups, with only 49.8% of B1 and 62.2% of B2 patients correctly classified. Notably, a significant proportion of misclassified B1 patients were categorised as B2 (83.2%), while misclassified B2 patients were often mistaken for B1 (57.0%) or C dogs (43.0%). It can be challenging to distinguish between B1 and B2 stages, as asymptomatic dogs with murmurs characterise both, and to differentiate between advanced B2 and C stages, where dogs adapt to cardiac enlargement to compensate for volume overload before developing congestive heart failure [9]. In such cases, echocardiographic and radiographic diagnostics are often necessary for differentiation [9,37]. The complexity of distinguishing between classes may have led to classification difficulties for the machine learning algorithm. A simplified model achieved a higher accuracy (90.8%) in classifying B dogs without distinguishing between B1 and B2. However, the ability to determine between stages B1 and B2 is essential in MMVD. Patients definitively diagnosed as B2 according to the ACVIM consensus guidelines [9] should begin chronic oral treatment with pimobendan in order to prolong the preclinical period and delay the onset of clinical signs [6].

The algorithm also had difficulty distinguishing between stages C and D in the complex model, with 87.5% of D dogs being misclassified as C. It is challenging to differentiate between stages C and D of cardiac disease in dogs, as clinical signs largely overlap. These clinical signs include tachypnoea, dyspnoea, hyporexia/anorexia, weight loss, and cough [9]. In the simplified model, both categories were grouped.

In general, while anamnesis and physical examination contribute to staging, especially for the control group and C dogs, that information could not distinguish B1 from B2 or C from D dogs. However, the simplified model could differentiate effectively between the control group, B, and C-D dogs. 

The simplified model could differentiate between the control group, B, and CD patients in the majority of dogs. With further development of the technique, this information could prove advantageous for general veterinarians who may be deficient in advanced cardiology expertise, as it may help them evaluate a patient with MMVD and determine the need for urgent referral to a cardiology centre. Furthermore, this tool could improve the cardiovascular evaluation of patients with MMVD without the risks of anaesthesia and sedation necessary for accurate ACVIM classification based on thoracic radiography [9] and, in certain geographic regions, could reduce radiation exposure to human operators.

An alternative set of simple tests has been suggested for diagnosing dogs with preclinical stage MMVD (B2), including biomarker-based diagnostics and tailored therapeutic management to avoid sedation for radiography [38]. Echocardiography, considered the “gold standard”, requires specialised training [9] and has limited accessibility in first-opinion practice. Although in this study, physical examination and history helped to classify a dog with MMVD, according to the ACVIM guidelines. It would be a mistake to start treatment without prior echocardiography. The importance of early detection should be prioritised for referral to a specialist for dogs suffering from MMVD. This is a fundamental part of the developed study because machine learning techniques do not intend to replace echocardiography and the ACVIM guidelines for diagnosis in dogs but may be useful as an additional tool in the disease classification, when further developed and studied.

Diagnostic methods based on artificial intelligence have recently been implemented in canine MMVD [21]. In the research developed by Valente et al. (2023) [21], the ability of machine learning techniques to assess the severity of MMVD from canine thoracic radiographs was investigated. Radiological studies of 1242 dogs in different phases of the disease were retrospectively analysed. The results in the study of the lateral radiological views showed an AUC of 0.87, 0.77, and 0.88 for stages B1, B2, and C+ D, respectively. The high accuracy of the algorithm in predicting the MMVD stage suggests that it could be a helpful support tool in the interpretation of canine thoracic radiographs. The previous artificial intelligence study determined that stage B1, C, and D dogs were better than stage B2 dogs. As in the conducted study, the worst classification results were obtained for the stage B2 dogs. The echocardiographic study and the radiological study of dogs in phase B2 are complex, with a great clinical variety among patients compared to animals classified as B1, C, or D.

As MMVD advances, FETCH-Q scores typically rise, and owners’ responses to questions during the anamnesis process also increase as clinical signs emerge or worsen in patients. These findings are consistent with previous studies [7,22]. The FETCH-Q survey contributes to dog classification to varying degrees. FETCH-Q question responses differ across the five ACVIM classes. The owner’s perception of their dog’s quality of life aligns with their assessment within the ACVIM classification, supporting our hypothesis. The responses obtained from the completion of a structured anamnesis through simple and objective questions by dog owners have been shown to be a critical factor in assessing cardiac diseases in primary veterinary care. The evaluation of the quality of life in dogs with MMVD is essential, and tools like the FETCH-Q scale can be beneficial. However, a conclusive diagnosis should consistently rely on support from imaging tests, particularly thoracic radiography and echocardiography [7,22]. 

The significance of an entire medical history is well known in human medicine. An adequate physical examination starts with a systematic patient history, which can improve diagnostic precision [39]. However, new technologies, online consultations, and artificial intelligence have disrupted this field [40,41]. 

Variances in clinical signs and physical examination outcomes across the groups were observed. As the ACVIM stage increased, the clinical signs became more severe. Hyporexia/anorexia and declining body condition were negative markers. These signs can indicate possible congestive heart failure [6,12,42]. Also, there was a correlation between poor outcomes and dyspnoea, cough, syncope, worsening body condition, and anorexia, as indicated by previous studies [11,38,42,43,44].

The physical examination remains fundamental in the ACVIM classification, distinguishing the control group from B dogs based on the presence or absence of a heart murmur [9]. A left apical systolic murmur is the initial clinical finding in MMVD patients [45,46]. The findings align with previous research, suggesting that murmur intensity beyond III/VI often coincides with advanced stages of heart disease and may indicate an increased risk of adverse outcomes [38,42,47].

The algorithm uses various physical examination variables to differentiate between stages, including heart rate, respiratory rate, and capillary refill time. Heart rate variations reflect how the sinus node responds to stimulation from the autonomic nervous system, which regulates heart rate through sympathetic tone. As MMVD severity increases, heart rate gradually rises, a trend confirmed by various studies [48,49,50], an observation confirmed in this study. Physical activity and environmental stress can temporarily increase heart rate [51].

Resting respiratory rate (RRR) is an easily measurable clinical parameter for clinicians and pet owners, highlighting its importance. Healthy dogs typically exhibit a RRR below 40 breaths per minute (bpm) in a clinical scenario [52,53,54] and below 30 bpm at home [53]. Our findings are consistent with previous studies indicating an increase in respiratory rate from stage B2 to stage C [50,53]. Capillary refill time (CRT) is a simple tool to assess peripheral perfusion, circulatory stability, and hydration status [54,55]. Our study found that CRT increased as MMVD progressed, which may indicate deteriorating haemodynamic [56]. In human medicine, it is commonly used to assess critically ill patients with cardiopulmonary pathologies [55]. CRT’s usefulness may be affected by inter-observer variability, and it is recommended to be used in conjunction with other diagnostic tests [56,57,58].

There were no significant differences in systolic blood pressure measurements between groups. Systemic hypertension can cause irreversible damage to organs, while systemic hypotension can result in circulatory shock due to inadequate tissue perfusion [59,60]. A decrease in systolic blood pressure with MMVD progression has been documented [6,60]. While the current study employed the same blood pressure measurement protocol, the absence of significant findings between groups could be attributed to the individual variability among dogs, the level of excitement or relaxation exhibited by patients, the ease of patient handling, and the relaxed and comfortable environmental conditions where measurements were conducted.

While the study conducted provides valuable insights, it also carries limitations. First, dogs were studied only once. Other studies follow up and observe variations over time [6,12], making their data more robust by following a repeated measures model. Second, despite the efforts made to establish consistent diagnostic criteria, the multicentre approach of the study resulted in some subjectivity in collecting data from the physical and echocardiographic examinations. Experienced veterinarians performed the echocardiographic measurements, which an ACVIM board-eligible professional reviewed. However, inter-observer variability in measurements may result in differences in specific measurements, which could affect the ACVIM classification of a dog [60,61] and, consequently, the accuracy of the machine learning algorithms. Third, the study included a diverse dog population with different breeds, ages, and sizes, which was essential for developing classification trees using machine learning techniques. Although many breeds are included in this study, it is worth noting the absence of dogs from breeds with a high predisposition to the development of MMVD, such as the CKCS [4], which could affect the results obtained. This may be due to the limited presence of CKCS in the countries where the study was conducted, which may be a limiting factor of this study. The inclusion of this breed in future studies may improve the results of our hypothesis. Concurrent diseases were not excluded, which mirrors the reality of geriatric patients with multiple comorbidities. Fourthly, owners’ subjective assessments may have been influenced by concomitant pathologies, his personal opinions and emotional state, and the dog’s clinical signs, making it challenging to differentiate cardiac-related signs. Similar observations have been seen in human patients with congenital heart disease and high anxiety levels [62] and in patients who withhold medical information due to embarrassment or ignorance [63,64]. The efficacy of machine learning techniques relies on proper pre-processing of data, precise training of models, and refinement of systems [65]. These factors pose significant challenges that require additional efforts to overcome. As such, exploring and implementing strategies that address these challenges are crucial to enhancing the accuracy and reliability of machine learning processes.

## 5. Conclusions

Machine learning techniques, based on a quality of life survey, clinical history, and physical examination, can be helpful additional tools when approaching dogs with MMVD in the first-opinion scenario. In most cases, the proposed model could classify healthy dogs and patients in stages B and C, according to the ACVIM classification for MMVD. However, it still faces difficulties differentiating between stages B1 and B2 and determining the advanced stages of the disease, C and D. Furthermore, the FETCH-Q survey showed that owners were aware of their dogs’ deterioration as MMVD progressed. To validate the algorithms, conducting clinical prospective studies on patients at different stages of MMVD would be necessary.

## Figures and Tables

**Figure 1 vetsci-11-00118-f001:**
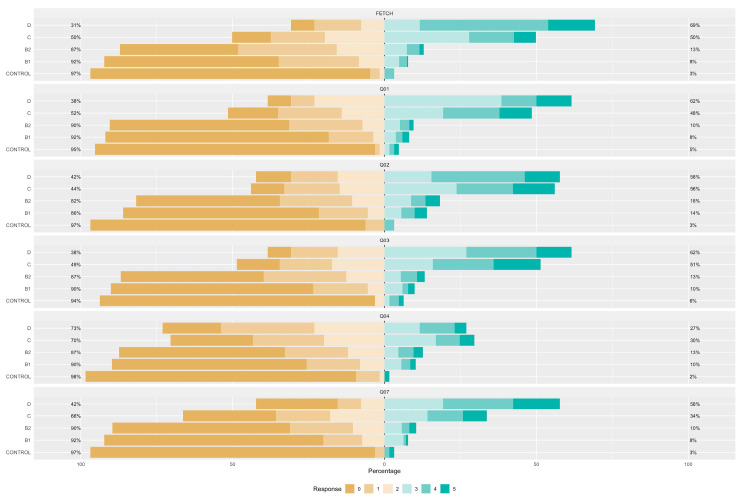
Likert representation of the answers for FETCH-Q and questions 1 to 7. Q1: “Does your dog have difficulty breathing?”; Q2: “Does your dog cough?”; Q3: “Does your dog often breathe very fast?”; Q4: “Does your dog snore when breathing?”; Q5: “Does your dog have difficulty in recreation? (Playing fetch, running, playing with other dogs or you, etc.)?”; Q6: “Were your dog’s favourite activities limited due to exercise restrictions by the veterinarian?”; Q7: “Does your dog sit or lie down during walks? (Does not tolerate exercise)”.

**Figure 2 vetsci-11-00118-f002:**
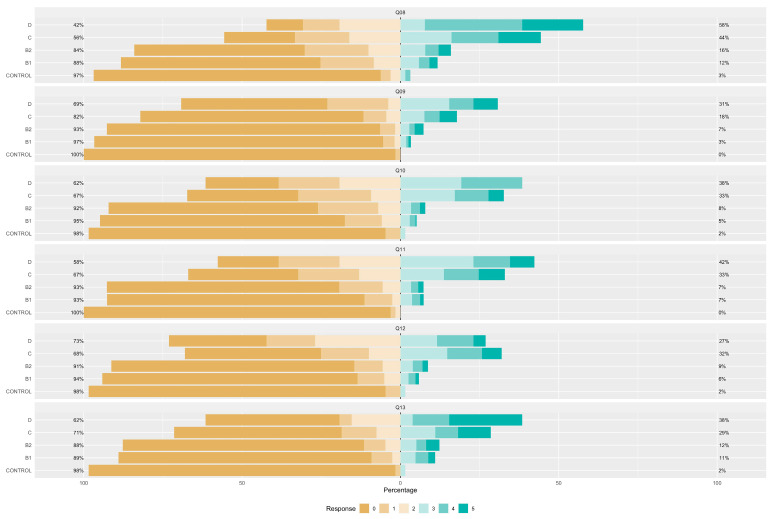
Likert representation of the answers for questions 8 to 13. Q8: “Does your dog have difficulty going up and down stairs?”; Q9: “Has your dog had episodes of collapse or fainting (syncope)?”; Q10: “Does your dog have difficulty getting comfortable? (At any time of the day).”; Q11: “Does your dog have difficulty sleeping through the night?”; Q12: “Is your dog eating less than he should, or has he been inappetent for the last few weeks?”; Q13: “Have you changed the type of food your dog is willing to eat?”.

**Figure 3 vetsci-11-00118-f003:**
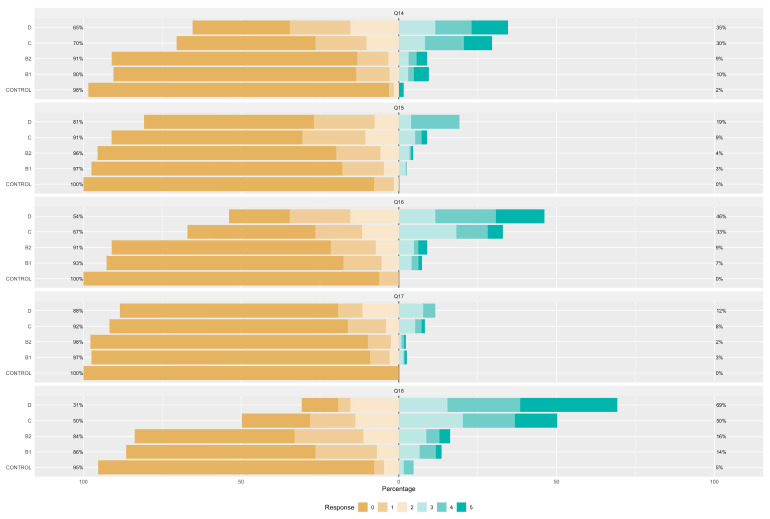
Likert representation of the answers for questions 14 to 18. Q14: “Increased urinary accidents in the house? (Urinating inside the house or where he should not)?”; Q15: “Has your dog had vomiting episodes?”; Q16: “? Has your dog had any limitations in spending time with you and the family (cannot get on the bed or sofa, avoids moving around, avoids bed or sofa, avoids moving)”; Q17: “Has your dog become irritable or unwilling to be touched? (Behaviour change)”; Q18: “Is your dog less active and vital?”.

**Figure 4 vetsci-11-00118-f004:**
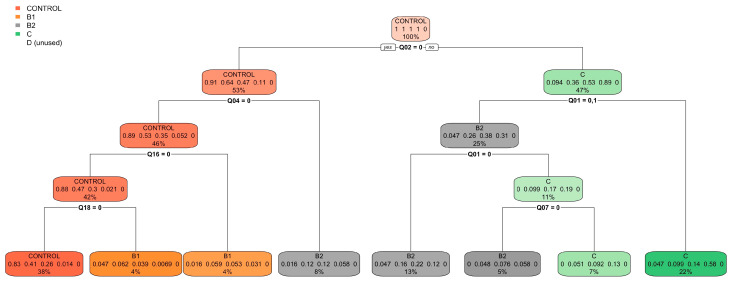
Classification tree for the model using the FETCH-Q. It represents the different selection criteria or ‘decision nodes’ used to predict the most correct classification of the total number of dogs (represented at the tree’s root as 100%). As the data are classified into subsets, the percentage value represents the probability of a dog belonging to that data subset. ACVIM class D was unused due to the few dogs recorded. Q1: “Does your dog have difficulty breathing?”; Q2: “Does your dog cough?”; Q4: “Does your dog snore when breathing?”; Q7: “Does your dog sit or lie down during walks? (Does not tolerate exercise).”; Q18: “Is your dog less active and vital?”.

**Figure 5 vetsci-11-00118-f005:**
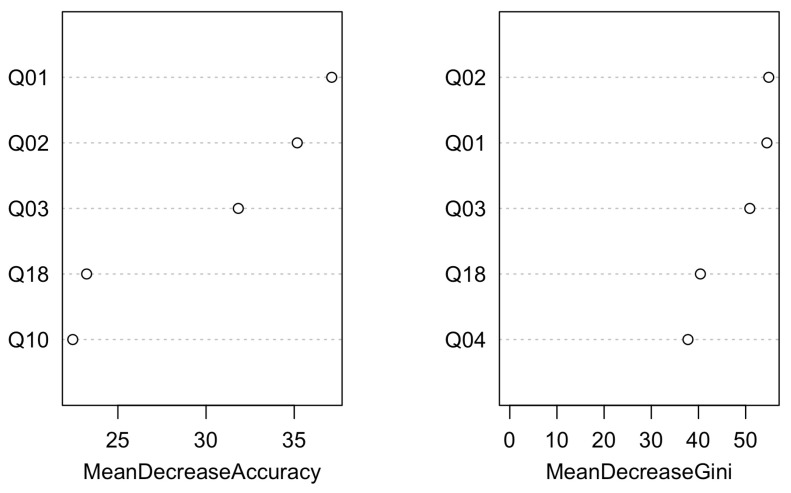
Variable importance plot of the first five variables for the model created using the FETCH-Q. The random forest algorithm measures the importance of each variable in classifying the data. The Mean Decrease Accuracy plot and Mean Decrease in the Gini coefficient help identify the variables that contribute most to the homogeneity of nodes and leaves in the forest. Variables are ranked in order of importance based on how much accuracy is lost when excluded. Q1: “Does your dog have difficulty breathing?”; Q2: “Does your dog cough?”; Q3: “Does your dog often breathe very fast?”; Q10: “Does your dog have difficulty getting comfortable? (At any time of the day)?”; Q18: “Is your dog less active and vital?”.

**Figure 6 vetsci-11-00118-f006:**
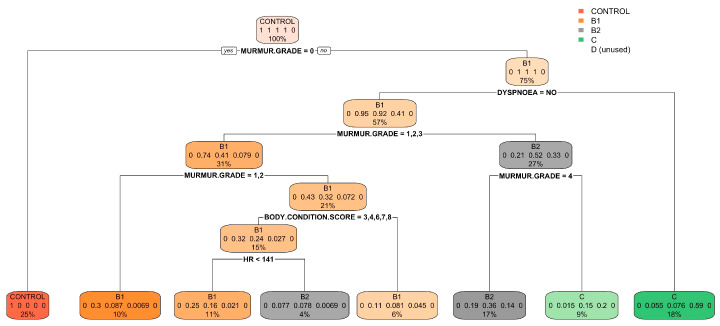
Classification tree for the anamnesis and the physical examination model. The classification tree represents the different selection criteria or ‘decision nodes’ used to predict the most correct classification of the total number of dogs (described at the tree’s root as 100%). As the data are classified into subsets, the percentage value represents the probability of a dog belonging to that data subset. ACVIM class D was unused due to the few dogs collected.

**Figure 7 vetsci-11-00118-f007:**
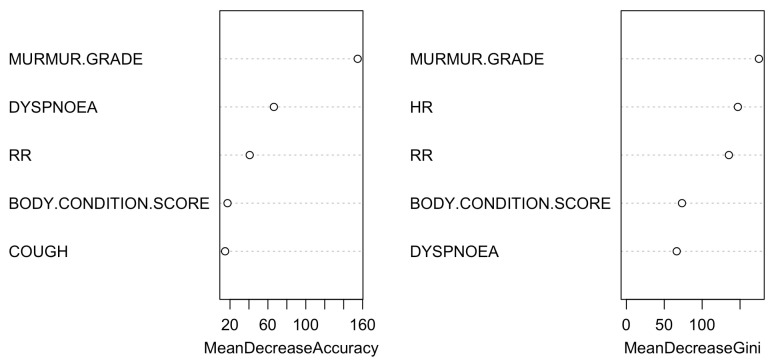
Variable importance plot of the first five variables for the model created using anamnesis and physical examination. The random forest algorithm measures the importance of each variable in classifying the data. The Mean Decrease Accuracy plot and Mean Decrease in the Gini coefficient help identify the variables that contribute most to the homogeneity of nodes and leaves in the forest. Variables are ranked in order of importance based on how much accuracy is lost when excluded. RR: respiratory rate; HR: heart rate.

**Figure 8 vetsci-11-00118-f008:**
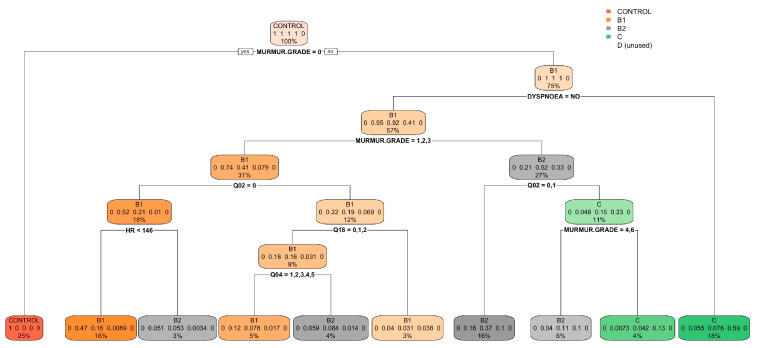
Classification tree for the model using the FETCH-Q, the structured anamnesis, and the physical examination. The classification tree represents the different selection criteria or ‘decision nodes’ used to predict the most correct classification of the total number of dogs (represented at the tree’s root as 100%). As the data are classified into subsets, the percentage value represents the probability of a dog belonging to that data subset. ACVIM class D was unused due to the few dogs collected. Q2: “Does your dog cough?”; Q4: “Does your dog snore when breathing?”; Q18: “Is your dog less active and vital?”.

**Figure 9 vetsci-11-00118-f009:**
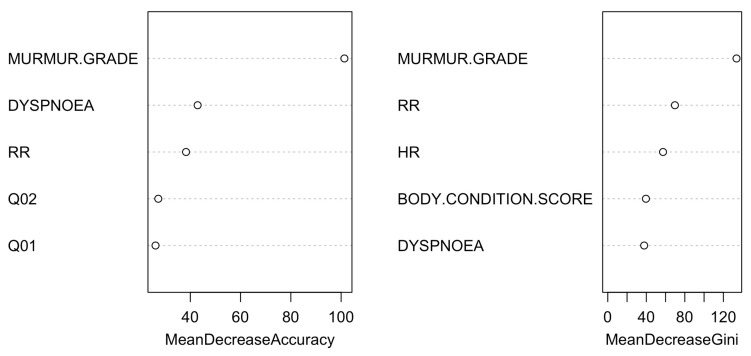
Variable importance plot of the first ten variables for the model created using the FETCH-Q, the structured anamnesis, and the physical examination. The random forest algorithm measures the importance of each variable in classifying the data. The Mean Decrease Accuracy plot and Mean Decrease in the Gini coefficient help identify the variables that contribute most to the homogeneity of nodes and leaves in the forest. Variables are ranked in order of importance based on how much accuracy is lost when excluded. RR: respiratory rate; HR: heart rate. Q1: “Does your dog have difficulty breathing?”; Q2: “Does your dog cough?”.

**Figure 10 vetsci-11-00118-f010:**
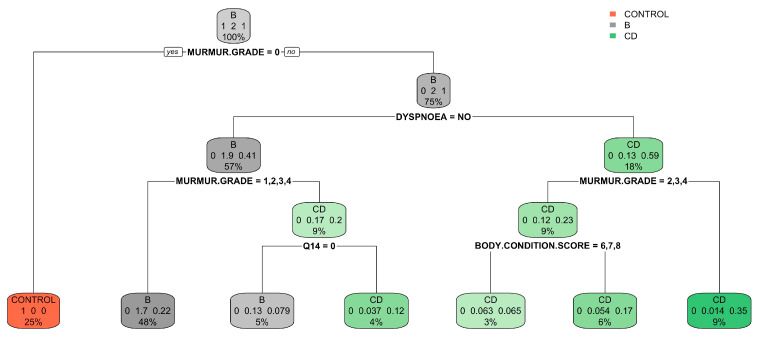
Classification tree for the simplified model using the FETCH-Q, the structured anamnesis, and the physical examination. The classification tree represents the different selection criteria or ‘decision nodes’ used to predict the most correct classification of the total number of dogs (represented at the tree’s root as 100%). As the data are classified into subsets, the percentage value represents the probability of a dog belonging to that data subset. Q14: “Increased urinary accidents in the house? (Urinating inside the house or where he should not)?”.

**Figure 11 vetsci-11-00118-f011:**
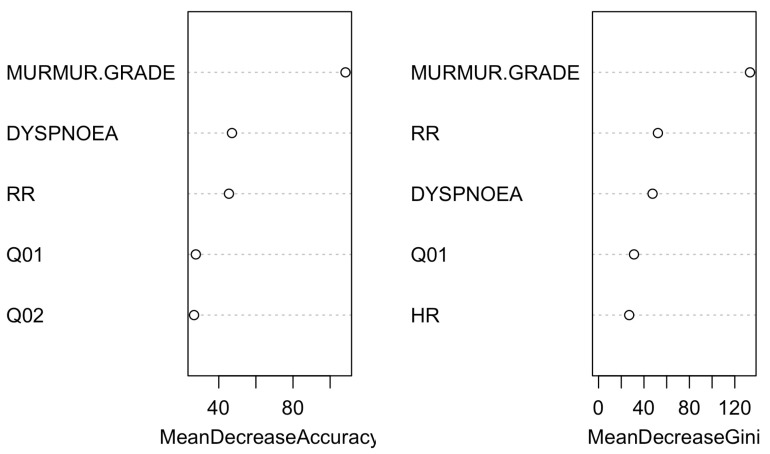
Variable importance plot of the first ten variables for the simplified model created using the FETCH-Q, the structured anamnesis, and the physical examination. The random forest algorithm measures the importance of each variable in classifying the data. The Mean Decrease Accuracy plot and Mean Decrease in the Gini coefficient help identify the variables that contribute most to the homogeneity of nodes and leaves in the forest. Variables are ranked in order of importance based on how much accuracy is lost when excluded. RR: respiratory rate; HR: heart rate; Q1: “Does your dog have difficulty breathing?”; Q2: “Does your dog cough?”.

**Table 1 vetsci-11-00118-t001:** Demographic data (sex, age, and body weight) of the control group (healthy dogs) and the MMVD groups (ACVIM B1, B2, C, and D).

		Control Group (N = 64)	B1 (N = 273)	B2 (N = 357)	C (N = 291)	D (N = 26)
**SEX**	**M**	**36 (56.3%)**	**139 (50.9%)**	**175(49%)**	**121 (41.6%)**	**11 (42.3%)**
	**F**	**28 (43.8%)**	**134 (49.2%)**	**182 (51%)**	**170 (58.4%)**	**15 (57.7%)**
**AGE (years)**						
**Median** **(Min, max)**		**6.0 (1.0, 16.0)**	**11.0 (2.0, 19.0)**	**12.0 (1.0, 18.0)**	**12.0 (5.0, 18.0)**	**12.8 (12.0, 17.0)**
**BODY WEIGHT (Kg)**						
**Median** **(Min, max)**		**13.2 (2.5, 48.5)**	**7 (1.0, 46.5)**	**7.5 (1.5, 47.5)**	**5.8 (1.5, 37.5)**	**12.8 (10.0, 17.0)**

M: male; F: female.

**Table 2 vetsci-11-00118-t002:** List of the more representative breeds in the control group (healthy dogs) and the MMVD groups (ACVIM B1, B2, C, and D).

	Control Group (N = 64)	B1 (N = 273)	B2 (N = 357)	C (N = 291)	D (N = 26)
**CROSSBREED**	**33 (51.6%)**	**101 (37.0%)**	**132 (37.0%)**	**93 (32.0%)**	**12 (46.2%)**
**BEAGLE**	**12 (18.8%)**	**8 (2.9%)**	**11 (3.1%)**	**8 (2.7%)**	**0 (0.0%)**
**YORKSHIRE TERRIER**	**5 (7.8%)**	**42 (15.4%)**	**40 (11.2%)**	**38 (13.1%)**	**3 (11.5%)**
**CHIHUAHUA**	**2 (3.1%)**	**23 (8.4%)**	**30 (8.4%)**	**50 (17.2%)**	**0 (0.0%)**
**MALTESE**	**0 (0.0%)**	**19 (7.0%)**	**14 (3.9%)**	**24 (8.2%)**	**2 (7.7%)**
**POODLE**	**0 (0.0%)**	**9 (3.3%)**	**20 (5.6%)**	**24 (8.2%)**	**2 (7.7%)**
**DACHSHUND**	**0 (0.0%)**	**11 (4.0%)**	**19 (5.3%)**	**10 (3.4%)**	**0 (0.0%)**
**MINIATURE** **SCHNAUZER**	**0 (0.0%)**	**8 (2.9%)**	**9 (2.5%)**	**7 (2.4%)**	**1 (3.8%)**
**SHIH TZU**	**0 (0.0%)**	**11 (4.0%)**	**7 (2.0%)**	**5 (1.7%)**	**0 (0.0%)**
**COCKER SPANIEL**	**0 (0.0%)**	**3 (1.1%)**	**12 (3.4%)**	**5 (1.7%)**	**0 (0.0%)**

**Table 3 vetsci-11-00118-t003:** Correlation between the clinical signs reported during history and the ACVIM classification.

		Control Group (N = 64)	B1 (N = 273)	B2 (N = 357)	C (N = 291)	D (N = 26)
**Cough**	Yes	**4 (6.3%) ^a^**	55 (20.1%) ^b^	94 (26.3%) ^b^	169 (58.1%) ^c^	17 (65.4%) ^c^
	No	60 (93,8%)	**218** (79.9%)	263 (73.7%)	122 (41.9%)	9 (34.6%)
**Dyspnoea**	Yes	**3 (4.7%) ^a^**	15 (5.5%) ^a^	27 (7.6%) ^a^	171 (58.8%) ^b^	16 (61.5%) ^b^
	No	**61 (95.3%)**	**258** (94.5%)	330 (92.4%)	120 (41.2%)	10 (38.5%)
**Syncope**	Yes	**1 (1.6%) ^ab^**	7 (2.6%) ^a^	30 (8.4%) ^b^	48 (16.5%) ^c^	7 (26.9%) ^c^
	No	**63 (98.4%)**	**266** (97.4%)	327 (91.6%)	243 (83.5%)	19 (73.1%)
**Exercise**	Yes	**1 (1.6%) ^a^**	**13 (4.8%) ^a^**	36 (10.1%) ^b^	100 (34.4%) ^c^	14 (53.8%) ^c^
**Intolerance**	No	**63 (98.4%)**	**260 (95.2%)**	321 (89.9%)	191 (65.6%)	12 (46.2%)
**Anorexia**	Yes	**0 (0%) ^a^**	**10 (3.7%) ^a^**	15 (4.2%) ^a^	42 (14.4%) ^b^	9 (34.6%) ^c^
	No	**64 (100%)**	**263 (96.3%)**	342 (95.8%)	249 (85.6%)	17 (65.4%)
**Body weight loss**	Yes	**0 (0%) ^a^**	5 (1.8%) ^a^	6 (1.7%) ^a^	19 (6.5%) ^b^	8 (30.8%) ^c^
	No	**64 (100%)**	**268** (98.2%)	351 (98.3%)	272 (93.5%)	18 (69.2%)

Data are presented as frequency tables. Classes with different letters are statistically different (*p* < 0.05).

**Table 4 vetsci-11-00118-t004:** Correlation between the physical examination and the ACVIM classification.

		Control Group (N = 64)	B1 (N = 273)	B2 (N = 357)	C (N = 291)	D (N = 26)
**Murmur**	Yes	0 (0%) ^a^	**273** (100%) ^b^	357 (100%) ^b^	291 (100%) ^b^	26 (100%) ^b^
	No	**64 (100%)**	0 (0%)	0 (0%)	0 (0%)	0 (0%)
**Murmur**	**No**	**64(100%) ^a^**	**0 (0%) ^b^**	**0 (0%) ^c^**	**0 (0%) ^d^**	**0 (0%) ^e^**
**grade**	**1**	**0 (0%)**	**18 (6.6%)**	**3 (0.8%)**	**0 (0%)**	**0 (0%)**
	**2**	**0 (0%)**	**71 (26%)**	**28 (7.8%)**	**4 (1.4%)**	**1 (3.8%)**
	**3**	**0 (0%)**	**124 (45.4%)**	**123 (34.5%)**	**30 (10.3%)**	**1 (3.8%)**
	**4**	**0 (0%)**	**56 (20.5%)**	**143 (40.1%)**	**97 (33.3%)**	**5 (19.2%)**
	**5**	**0 (0%)**	**4 (1.5%)**	**52 (14.6%)**	**141 (48.5%)**	**12 (46.2%)**
	**6**	**0 (0%)**	**0 (0%)**	**8 (2.2%)**	**19 (6.5%)**	**7 (26.9%)**
**CRT**	>2 s	0 (0%) ^a^	2 (0.7%) ^a^	1 (0.3%) ^a^	5 (1.7%) ^a^	3 (11.5%) ^b^
	<2 s	**64 (100%)**	**271** (99.3%)	356 (99.7%)	286 (98.3%)	23 (88.5%)
**HR**	bpm	**107** [60, 176] ^a^	120 [60, 220] ^b^	124 [55, 230] ^b^	142 [60.0, 290] ^c^	150 [85.0, 260] ^c^
**RR**	bpm	24.0 [12, 60.0] ^a^	24.0 [15, 100] ^a^	24.0 [12, 90.0] ^a^	44.0 [16.0, 180] ^b^	40.0 [24.0, 210] ^b^
**SAP**	mm Hg	136 [101, 187] ^a^	134 [73, 206] ^a^	130 [79, 224] ^a^	140 [75, 210] ^a^	135 [95, 154] ^a^
**DAP**	mm Hg	**84.0** [48, 158] ^a^	88.0 [48, 142] ^a^	87.0 [51, 150] ^a^	89.0 [36, 129] ^a^	92.0 [60, 113] ^a^
**MAP**	mm Hg	**95.0** [65, 163] ^a^	98.0 [59, 147] ^a^	95.0 [70, 166] ^a^	98.0 [57, 148] ^a^	107 [71, 120] ^a^
**RT**	°C	38.0 [37.0, 40.0] ^a^	38.1 [35.4, 40.5] ^a^	38.2 [36.7, 39.7] ^a^	38.2 [35.0, 39.5] ^a^	38.1 [36.6, 39.3] ^a^

Data are presented as frequency tables and medians [minimum, maximum]. Classes with different letters are statistically different (*p* < 0.05). CRT: capillary refill time. HR: heart rate. RR: respiratory rate. SAP: systolic arterial pressure. DAP: diastolic arterial pressure. MAP: mean arterial pressure. RT: rectal temperature.

**Table 5 vetsci-11-00118-t005:** Classification matrixes of the complex models (echo-based classification). They illustrate the accuracy of the models in correctly classifying dogs based on the ACVIM classification criteria. Each row corresponds to an ACVIM class and displays the number of dogs classified into various categories by the model. The ‘class error’ represents the percentage of misclassified dogs when utilising the model.

**FETCH-Q model**						
	**Control group**	**B1**	**B2**	**C**	**D**	**Class error**
**Control group**	**2**	**56**	**3**	**3**	**0**	**97%**
**B1**	**2**	**143**	**84**	**44**	**0**	**48%**
**B2**	**0**	**126**	**138**	**93**	**0**	**61%**
**C**	**0**	**14**	**72**	**204**	**0**	**30%**
**D**	**0**	**0**	**3**	**21**	**2**	**92%**
**Physical examination model**						
	**Control group**	**B1**	**B2**	**C**	**D**	**Class error**
**Control group**	**64**	**0**	**0**	**0**	**0**	**0%**
**B1**	**0**	**149**	**111**	**13**	**0**	**45%**
**B2**	**0**	**106**	**189**	**62**	**0**	**47%**
**C**	**0**	**20**	**68**	**201**	**2**	**31%**
**D**	**0**	**3**	**5**	**15**	**3**	**88%**
**FETCH-Q plus physical examination model**						
	**Control group**	**B1**	**B2**	**C**	**D**	**Class error**
**Control group**	**62**	**1**	**0**	**1**	**0**	**3%**
**B1**	**0**	**136**	**114**	**23**	**0**	**50%**
**B2**	**0**	**77**	**222**	**58**	**0**	**38%**
**C**	**0**	**9**	**57**	**224**	**0**	**23%**
**D**	**0**	**1**	**2**	**21**	**2**	**92%**

**Table 6 vetsci-11-00118-t006:** Classification matrix of the simplified model (echo-based classification). Categories B1 and B2 are grouped into category B, and categories C and D are grouped into category CD. The ‘class error’ represents the percentage of misclassified dogs when utilising the model.

Simplified Model				
	**Control Group**	**B**	**CD**	**Class Error**
**Control group**	**60**	**3**	**1**	**6%**
**B**	**0**	**572**	**58**	**9%**
**CD**	**0**	**84**	**232**	**27%**

## Data Availability

Data supporting the reported results can be sent to anyone interested by contacting the corresponding author.

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
