# Peer review of "Machine Learning Techniques for Canine Myxomatous Mitral Valve Disease Classification: Integrating Anamnesis, Quality of Life Survey, and Physical Examination"

_vetsci, 2024, doi:10.3390/vetsci11030118_

Round 1
Reviewer 1 Report
Comments and Suggestions for Authors
This is an interesting study investigating the potential of Machine Learning Techniques as an aid in staging dogs with myxomatous mitral valve disease (MMVD); an area with potential future implementation of clinical value. However, there are several important issues that need clarification and modification.
General comments:
An important issue to clarify is whether included dogs all underwent the three parts included in the study design, i.e. Anamnesis, the Quality of Life Survey, and the Physical Examination, as well as echocardiography for diagnosis of MMVD. The information differs between M&M where it says OR (line 125) and for example the abstract and other parts of the manuscript. This needs to be clarified throughout the manuscript. To me, it should be mandatory, i.e. an inclusion criteria, that all dogs have undergone these 4 parts. And of course it is important that it is the same dogs taking part in all 4 parts.
I strongly recommend using the term myxomatous mitral valve disease (MMVD), which is a more correct pathophysiological term for the disease.
It needs to be clarified if all dogs (except controls) were diagnosed with MMVD using echocardiography? What were the diagnostic criteria for the diagnosis? Were dogs with other cardiac diseases excluded? The inclusion and exclusion criteria need to be better described.
It also needs to be clarified that you are not assessing the utility of the machine learning techniques for diagnosis of MMVD, but for staging of the disease, right? That is: you already know that all dogs (cases) have MMVD (diagnosed by echocardiography). If this is not the case, please clarify and explain.
Throughout the manuscript, make sure to differentiate from which of the 3 parts (Anamnesis, Survey or Physical Examination) your data is taken, eg when describing clinical signs (please also use the term clinical signs throughout, not symptoms).
It is important to state whether the examiners were aware of the status of the dog (echocardiographic results/diagnosis/stage of MMVD) when performing anamnesis and physical examination, if they were, there is a risk of influencing the results of eg cardiac auscultation. In the same manner: were examiners performing echocardiography aware of information from anamnesis and phys examination or were they blinded? It is also important to state if owners were aware of the echocardiographic diagnosis and staging when responding to the survey, as this could also affect responses. This needs to be clearly described and discussed in the manuscript.
The section describing hypotheses and aims (67-82) needs to be rewritten in a more straightforward way (avoid the wording In summary here)
It needs to be clarified if the dogs assessed as ACVIM stage A are the ones you describe as your controls or were the controls other dogs?
You state using a P-value<0.05 for statistical significance, which is fine as a starting point. However, you perform several tests between a number of groups, eg between 5 groups when using the ACVIM classification. In these circumstances, you need to correct the P-value for multiple comparisons, by eg. Bonferroni adjustment (which would give you an adjusted P-value of <0.005). Please adjust. Also go through the manuscript correcting text adhering to significant differences as a number of these are likely to change with a stricter P-value.
I was surprised to see the age of included dogs starting at 0.5 years: I assume this must have been a control dog? I would advise to only include adult dogs, i.e. with an age limit of eg. above 1y of age.
Generally, in the M&M and results, it needs to be clearer which dogs are cases and which dogs are controls.
In the Results section, the characterisation of the study population needs to be expanded to also specify age, BW and breeds for cases versus controls, to better describe the study population.
Your machine learning model shows some promising results. However, if I understand correctly (table 4) 82 of the CD dogs were classified as B, i.e. 26% were misclassified. That is more than 1 dog in 4. I do not see any information on whether these dogs were all stage C which had been stabilised on medication (please correct me if I’ve missed that information). Hence you cannot claim that your model can identify stable and unstable MMVD and furthermore not with ‘minimal data’ (you use quite a lot of data in your model), as written in eg. the abstract. This needs to be rephrased. Furthermore, it should be discussed that treatment (pimobendan) is recommended for stage B2 dogs; hence separation between B1 and B2 is of clinical importance.
Major comments:
Abstract:
Please rephrase abstract based on the comments given in this review. Especially the conclusion needs rephrasing.
Also, line 26: use disease instead of condition
Introduction:
As written above, the section describing hypotheses and aims (67-82) needs to be rewritten. Here also please rephrase: you cannot test prediction of the diagnosis of MMVD, but the classification/staging of dogs. For prediction of diagnosis you would need a variety of diagnoses (made by echocardiography) to test your machine learning variables (anamnesis, survey, physical ex) against.
Line 45 please specify: ‘it accounts for up to 75% of cases’: which cases?
Please emphasize that the actual diagnosis of MMVD is made by echocardiography (lines 50-56).
Line 63: offering flexibility? please explain
Line 70-71 change to: One study has shown (one reference)
M&M:
Line 87: please specify postgraduate training
Line 95: please specify structural or functional abnormalities: cardiac? other? Existence of heart murmurs? Overall, assessment and testing of controls need to be clarified.
Line 96: written consent?
Lines 109-110: please specify extensive experience
Were specific standardised protocols used for structured anamnesis and physical examinations? Were scales used for answers, eg. 0 1 2 3? Were answer alternatives Yes/no? Specify if protocols were all in Spanish? These protocols should preferably be added as supplementary material. (if in Spanish, they should be translated to English). If, not please give detailed description on how instructions were given to examiners and minimal examinations made.
The so called comprehensive physical examination should be better described.
Diagnostic criteria for the diagnosis of MMVD need to be better described.
The section describing the echocardiographic examination should be restructured starting with all information on echo machines etc followed by acquisitions used/ measurements performed.
Statistical analyses:
You state using a P-value<0.05 for statistical significance, which is fine as a starting point. Hoqwever, you perform several tests between a number of groups, eg between 5 groups when using the ACVIm classification. in these circumstances, you need to correct the P-value for multiple comparisons, by eg. Bonferroni adjustment (which would give you an adjusted P-value of <0.005). Please adjust.
Results:
Generally, it needs to be clearer which dogs are cases and which dogs are controls.
The characterisation of the study population needs to be expanded to also specify age, BW and breeds for cases versus controls. It needs to be clarified if the controls = ACVIM stage A or if the controls were other dogs? I was surprised to see the age starting at 0.5 years: I assume this must have been a control dog? If not, MMVD would be extremely unusual at this early age (must not be confused with mitral dysplasia), please check. I would advise to only include adult dogs, i.e. with an age limit of eg. 1y of age.
It is unclear whether the number you indicate as participants contributed to all four parts of your study, i.e all four examinations (which would be recommended). If not, the number of dogs participating/contributing to the different parts need to be specified.
Results re given from some methods are not described in M&M, such as BP measurement, please amend.
Discussion:
Overall, the discussion needs to be modified according to above given comments.
First section: to enhance you would have needed to compare your results with an assessment/statement given by the veterinarian for each dog. Was this done, ie. was the veterinarian asked to stage the dog based on anamnesis and physical examination (and potentially survey) results? Please clarify.
Again, 1 of 4 CD dogs were misclassified, please rephrase from line 317. As written above, I do not see any information on whether these dogs were all stage C which had been stabilised on medication. Please expand on this discussion. Furthermore, it should be discussed that treatment (pimobendan) is recommended for stage B2 dogs; hence separation between B1 and B2 is of clinical importance.
Further comments on discussion can be given in the next version.
Figures & Tables:
Generally, the font size of the figures is too small, please amend for better readability. Also, in the figures, specify nr of responses when appropriate
Fig 1: please explain Fetch Q total. Further, I do not see Q 5 and 6 in the graph, please modify.
In the figures with classification trees, please expand figure legend explaining the figure, especially specify what . 25 .25 .25 etc means.
Fig %: explain what you mean by first ten variables = ten most important?
Table 1: it is important that it is clear from what examination/part the clinical signs depicted are taken, eg. cough (survey, anamnesis, physical ex?). Please clarify
Table 2: check line Murmur grade: no letters a b c d e ?
Table 4: clarify which is echo-based classification.
Minor comments:
Keywords:
Please check journal instructions regarding inclusion of keywords already given in the title
References
Computer programs etc are usually not given in reference list, but instead as footnotes, please check journal instructions.
Some of the references, especially from text books are pretty old, please check and replace or complement with updated/newer references where appropriate.
Avoid ‘In summary’ (line 335) in the middle of discussion, save for conclusion!
Avoid using bpm for RR, usually reads beats per minute and used for HR
Comments on the Quality of English LanguageAcceptable quality.
Author Response
Document with comments and revisions is attached. Thank you.

Reviewer 2 Report
Comments and Suggestions for Authors
The value of the proposed article lies in the fact that, at a time when the scientific community is more strongly needed, more accurate instrumental quantification criteria of mitral insufficiency than those proposed by the ACVIM consensus in order to allow specialists to more adequately document when the right time is to start a medical therapy. This article responds to an equally important need, especially for general clinicians. This study confirms that an accurate evaluation according to a simple clinical algorithm such as FETCH-Q, which has no operating costs, can allow the practical clinician to improve the appropriateness of the indication to perform more expensive instrumental examinations. In a global clinical approach, these results, combined with those of instrumental clinical research, will improve the real clinical usability of the classification established by the ACVIM consensus.
In the introduction, the authors state (line 47-48) that “Early diagnosis and staging of this condition are critical to slow its progression and improve patient outcomes” here or in the discussion I would suggest to the authors to emphasize that this is fundamental to help to decide the moment and opportunity of a possible therapeutic intervention.
In many points of the article, it is stated that it is impossible, with this method, to distinguish between B1 and B2 this is obvious, and this result was imaginable given that in order to evaluate a remodeling in the compensated phase (B2) a cardiovascular imaging study is indispensable. However, this is the crucial point of the study, which if on the one hand shows that with a clinical examination and with an interview with an owner, we can more appropriately indicate an examination on the other hand shows that it would not be correct to give therapeutic indications without an instrumental examination, this should be emphasized in the discussion.
The absence of Cavalier King Charles Spaniels among the most represented breeds seems strange, this could depend on the lesser diffusion of these breeds in the countries where the study was carried out. However, it is factual that this decreases the value of the study, in fact in this breed it would seem that there is a difference between the clinical symptoms and therefore the data proposed in the FETCH-Q the degree of remodeling of the left heart chambers, compared to other breeds. This aspect could be described within the limits of the study and could be a study hypothesis for further research on the topic.
Furthermore, I recommend the authors to review the typos on line 310 there is an inverted question mark, used only in written Spanish.
At lines 340-341 it is stated “This tool could help first-opinion clinicians 340 assess cardiac urgency without the risks of anesthesia and sedation required for precise ACVIM classification based on thoracic radiography”
This is only true for some countries where animals are sedated to take an X-ray, in others, animals are held in place by veterinarians or radiology technicians, this could be clarified and stating that this reduces radiation exposure for human operators, this point could be emphasized.
Author Response
Attached is the document with comments and revisions. Thank you.

Round 2
Reviewer 1 Report
Comments and Suggestions for Authors
The manuscript has improved and is now much clearer and easier to follow. Many of my points have been answered and clarified. However, there are still some major points, and a number of other points, that need to be answered and ameliorated.
Major points:
11. The use of the term Stage A for the controls in the current manuscript is not correct. According to the Consensus statement for MMVD (Keene et al JVIM 2019): “Stage A identifies dogs at high risk for developing heart disease but that currently have no identifiable structural disorder of the heart (eg, every Cavalier King Charles Spaniel or other predisposed breed without a heart murmur)”. In your M&M you write: The control group were animals evaluated prior to elective surgery and were established as ACVIM A stage dogs based on the fact that they were healthy dogs at the time of evaluation; did not present clinical signs (absence of cardiorespiratory symptoms and heart murmur) and did not receive any medication.
This is fine for a control group (however add a comment on signs of other diseases as well, please see below) but it does not automatically make all these dogs MMVD Stage A dogs. From Table 1, it can be read that the BW for controls was: 13.2 (2.5,48.5) kg. Dogs over 40 kg in BW would not normally belong to a breed at high risk for developing MMVD. Furthermore, many of your so called Stage A dogs were crossbreds, ie. it would be difficult to tell whether they belong to a predisposed breed, right? The easiest solution for this would be to remove the term Stage A entirely from the manuscript, replacing it with Controls/Control group instead (the study would still hold interesting results). If you want to keep the term Stage A you will have to go through all controls, checking that they fit into the definition above, and excluding all dogs which do not. Again, I think that would be difficult with such a high number of crossbreds, and doing so would probably substantially reduce the number of dogs in this group. Obviously, all statistical analyses would have to be rerun as well.
22. As pointed out also in the previous review, it is essential that you do not claim that you can diagnose any heart disease, including MMVD, by survey, anamnesis and or physical examination. The text has been improved in some parts, but the problem partly still remains, such as in line 47: “This machine-learning approach provided adequate findings and was shown as an alternative additional tool for the diagnosis of MMVD” and in Line 413: “A structured anamnesis with straightforward, objective questions, such as the FETCH-Q quality-of-life survey, should be essential in diagnosing heart disease in first-opinion veterinary practice.” Please rephrase, and go through the ENTIRE manuscript checking this.
33. It is important to discuss that treatment (pimobendan) is recommended for stage B2 dogs; hence separation between B1 and B2 is of clinical importance. Although pointed out during my previous review, this point has not been sufficiently clarified, please add in discussion section, preferably in the second paragraph where stages B1 and B2 are discussed.
44. Upon finalization of the manuscript, proof-reading by a native English-speaking professional is strongly recommended.
Other points:
Line 30: replace starting treatment early by: to evaluate when treatment should be initiated
Line 42: remove the word complete
Line 44: what do you mean by robust performance. This could be omitted in my opinion.
Line 48: replace diagnosis by staging, and see above regarding the term stage A
Line 60: MMVD avoid starting sentence with abbreviation
Line 63: what does Id mean?
Line 68: However, classifying patients solely based on medical history and clinical signs can be challenging: please rephrase, it is not possible to classify patients solely based on medical history and clinical signs.
Line 74: again, you cannot diagnose MMVD with your tool, but it may aid in classifying dogs
Line 82: replace research by study
Line 89-90: “correlation between owners’ perception and ACVIM classification for MMVD even when owners were unaware of the specific cause of their pets’ clinical signs” Was this correlation evaluated? Please clarify
Lines 91-97 from This innovative… please move entire text to discussion
line 104: please explain IVPSP
line 105. were cardiology residents allowed to make evaluations or were they supervised by their supervisors, please clarify
Line 121 regarding controls: only free of cardiorespiratory signs, or also of signs of other diseases? (systemic or organ-related diseases). Please clarify. Please also use the term clinical signs instead of symptoms, as previously commented
Line 134: did the sonographers also lack other information on other parts of the dogs, such as anamnesis and clinical signs? please clarify
Line 197: please use the wiord dog for controls (not case)
Please use the term body weight instead of weight, throughout.
Table 2 check spelling of breeds
Some of the Figures (Fig 1-3, 6-7): font size still too small, please rectify
Line 208: what do you mean by differences here? significant differences? please clarify
Line 209: what do you mean by indicated here? please clarify
Fig 5: If possible, please use a program for figures/graphs to improve appearance, or at least please omit the word FOREST above the figure and improve the text under the X-axes.
Fig 7 and 11: If possible, please use a program for figures/graphs to improve appearance, or at least please omit the word FOREST above the figure and improve the outline of all parts including written text.
Table 3. Relationship between the clinical signs: specify if clinical signs have been reported during history or observed during physical examination.
Also, Line 233: clarify which table is which: Tables 3 and 4 provide insights into clinical signs and physical examination
Tables 3 and 4: check that the heading really explains what is seen in the table; preferably exchange the word Relationship here
Table 5. Again, could you please clarify in the table which is echo-based classification. Also the headings (models) all need to be centered (FetchQ not centered)
Table 6. Same as above
Fig 10. unclear image, please amend
Discussion:
Line 349: please rephrase to: presence of a heart murmur and/or cardiorespiratory signs
Line 352 please rephrase to: can aid in the initial assessment…
Line 352 please rephrase to: accurately identify the B groups,
Line 363 please rephrase to: echocardiographic and radiographic diagnostics
Line 383 and following: Please make sure the reference is correctly used here and that this sentence is correct: An alternative set of simple tests is suggested for diagnosing dogs with preclinical stage MMVD (B2), followed by biomarker-based diagnostics and tailored therapeutic management to avoid sedation for radiography has been proposed[38].
Also (the same sentence), please rephrase to: Has been suggested for identifying dogs with preclinical… and rephrase to: including biomarker-based diagnostics
Line 388 please omit identify (you did not identify MMVD)
Line 393: what do you base this on: “and is emerging as an simple effective additional tool”. ? Please omit. You can replace by: … but may be useful as an additional tool in classification of dogs, when further developed and studied.
Line 407: check English, please rephrase: the owner’s responses to questions during anamnesis worsen.
Line 413: please rephrase, again you cannot DIAGNOSE MMVD or other heart diseases using a survey: essential in diagnosing heart disease
Line 454: please specify if you did not use a standardised protocol for BP measurenments in the study: However, the study's absence of significant findings could be attributed to the variability in blood pressure measurement protocols, as routine blood pressure assessment is not universally practised in veterinary consultations.
line 475: please explain what you mean by: which required further investigation into potential interference with study outcomes

Upon finalization of the manuscript, proof-reading by a native English-speaking professional is strongly recommended.
Author Response
We sincerely value your comments and suggestions. The document has been rectified, and the corrections are indicated in blue. Additionally, I have enclosed the responses to your recommendations.

Round 3
Reviewer 1 Report
Comments and Suggestions for Authors
The manuscript is improved, and I congratulate the authors on the efforts made. I do, however, still have some points that need further improvement, two of which are major:
1. As pointed out already in my first review of this manuscript, over a quarter of CD dogs would be classified as B dogs (table 6, 27% class error in classification of CD dogs), meaning that a substantial number of CD dogs would be misclassified as B dogs and thus not referred to a specialist. This needs to be clearer in the text, especially in the abstract and in the main conclusions of the study, please see specific comments on the wording in these two parts below.
2. In conjunction with the above comment, you write that you classify the MMVD dogs according to the ACVIM guidelines, but you do not mention radiographic examination of dogs staged C or D. Was this examination performed to verify the stage of the disease, as advised in the guidelines? If not, this needs to be discussed and added as a major study limitation.
Simple summary: The text looks fine now, but please replace animal by dog throughout.
Abstract:
First sentence; add the classification already here to give background to the reader:
Myxomatous Mitral Valve Disease (MMVD) is a prevalent canine cardiac disease typically diagnosed and classified using echocardiography.
As we have discussed previously you are not assessing the technique for diagnosis of MMVD, and this should therefore not be included in the conclusion of the abstract. Furthermore, I find the term ‘high efficiency’ misleading. As pointed out above, as well as in the first review, there is a comparably large percentage of misclassification of CD patients, which has to be clear in the text. And again, you should not use the term case for the control dogs. Please replace the text from line 44 with the following:
Employing a classification tree and a random forest analysis, the complex model accurately identified 96.9% of control group dogs, 49.8% of B1, 62.2% of B2, 77.2% of C, and 7.7% of D cases. To enhance clinical utility, a simplified model grouping B1 and B2, and C and D into categories B and CD improved accuracy rates to 90.8% for stage B, 73.4% for stages CD and 93.8% for the control group. In conclusion, the current machine-learning technique was able to stage healthy dogs and dogs with MMVD classified into stages B and CD in the majority of dogs using quality of life surveys, medical history, and physical examinations. However, the technique faces difficulties differentiating between stages B1 and B2, and determining between advanced stages of the disease.
And in the main conclusion, you need to remove the following sentence:
Nevertheless, it can effectively distinguish between patients who require urgent assessment by a cardiology specialist.
Introduction:
Line 85: Valente et al, like yourselves, did not use the technique for diagnosis but for classification, please rephrase to:
In addition, a recent study has shown the ability of machine learning techniques to classify patients affected by MMVD using thoracic radiographs [21].
Line 88: please replace focus by aim, and research by study
Line 117: please add: left-sided systolic heart murmur
Lines 126 and 128: heart murmur
Line 132: replace by: echocardiographic examination
Line 142: please replace validated in the… by: according to ACVIM guidelines
Line 192: replace cases by dogs
Table 1: add units for age and body weight
Line 204: please replace: as the ACVIM classification declines, by: with increasing severity according to ACVIM classification
Font size and text appearance in figures: I leave the decision to the editor.
Line 228: replace by: physical examination findings
Discussion, first section:
Please remove the new text here, based on above major comments; thus remove: ‘This innovative approach has the future potential to improve the early detection and treatment of MMVD, offering a valuable tool for rapid decision making, particularly in situations where imaging techniques are not available.’
Also remove: ‘Ultimately, implementation of this approach could significantly improve the overall health and well-being of canine patients with MMVD.’
Please also modify the sentence beginning on line 354 slightly into: Furthermore, the technique allowed capturing the owner's perspective on disease progression, contributing to a comprehensive understanding of the disease [16].
Line 359: please rephrase to: The complex model correctly classified control dogs and stage C patients, achieving 96.9% and 77.2% accuracy, respectively.
Line 372: please rephrase to: Patients definitively diagnosed as B2 using the ACVIM guidelines (include the reference) should begin chronic oral treatment with pimobendan, in order to prolong the preclinical period and delay onset of clinical signs [change to original reference, your nr 6, Boswood et al 2016)
Line 383: please rephrase to: “The simplified model could differentiate between the control group, B and CD patients in the majority of dogs. With further development of the technique”…, this information could prove…
Line 412: please replace research by study
Line 423: please replace diagnosing by evaluation or assessing
Study limitations:
As written above, under point 2, it is important to clarify whether radiographic examination of dogs in stage C and D was performed and if not this should be added as a first major study limitation in the section of study limitations.
Line 507: please replace are by were

Author Response
Please, see the attachment.
